# Enhancing prosthetic vision by upgrade of a subretinal photovoltaic implant in situ

Mohajeet B. Bhuckory [1,2] ✉, Nicharee Monkongpitukkul[2,3,7], Andrew Shin[4,7], Anna Kochnev Goldstein [5], Nathan Jensen [5], Sarthak V. Shah [2], Davis Pham-Howard [1,2], Emma Butt [6], Roopa Dalal[2], Ludwig Galambos[1], Keith Mathieson [6], Theodore Kamins [5] & Daniel Palanker [1,2]

In patients with atrophic age-related macular degeneration, subretinal photovoltaic implant (PRIMA) provided visual acuity up to 20/440, matching its 100 μm pixels size. Next-generation implants with smaller pixels should significantly improve the acuity. This study in rats evaluates removal of a subretinal implant, replacement with a newer device, and the resulting grating acuity in-vivo. Six weeks after the initial implantation with planar and 3-dimensional devices, the retina was re-detached, and the devices were successfully removed. Histology demonstrated a preserved inner nuclear layer. Re-implantation of new devices into the same location demonstrated retinal re-attachment to a new implant. New devices with 22 μm pixels increased the grating acuity from the 100 μm capability of PRIMA implants to 28 μm, reaching the limit of natural resolution in rats. Reimplanted devices exhibited the same stimulation threshold as for the first implantation of the same implants in a control group. This study demonstrates the feasibility of safely upgrading the subretinal photovoltaic implants to improve prosthetic visual acuity.

Loss of photoreceptors in inherited retinal diseases or age-related macular degeneration (AMD) are among the leading causes for irreversible blindness. The inner retinal neurons remain to a large extent[1–3], albeit with some level of retinal circuits remodeling[4,5]. We developed a wireless photovoltaic subretinal implant capable of eliciting visual percepts through electrical stimulation of the second order neurons, bipolar cells (BC), originally receiving an input from photoreceptors[6,7]. A camera on augmented reality glasses captures the surrounding visual scene at eye level, and processed images are projected onto the implant using pulsed near infrared (NIR, 880 nm) light. This design confers several advantages compared to wired implants, including: relative ease of implantation, reduced risk of post-operative complications, the projected NIR does not interfere with the remaining peripheral natural vision, network mediated retinal stimulation

(through bipolar cells) preserves many features of the natural signal encoding in the retina, and others[6,8].

This approach has been tested in clinical trials with AMD patients, where PRIMA implants (Pixium Vision SA, Paris, France) with 100 μm pixels were implanted in the area of geographic atrophy. These patients, previously with no foveal light perception, gained monochromatic formed vision with a prosthetic visual acuity ranging from 20/438 to 20/550[9,10], closely matching the limit of this pixel size (20/420). The patients simultaneously perceive central prosthetic and natural peripheral vision[10]. While these landmark results are an important proof of concept for our technology, the number of AMD patients who would benefit from this technology would grow significantly if the implant resolution would be increased. For an acuity of 20/200 or better, pixels of 50 μm or smaller would be needed.

[1]Hansen Experimental Physics Laboratory, Stanford University, Stanford, CA 94303, USA. [2]Department of Ophthalmology, Stanford University, Stanford, CA, USA. [3]Department of Ophthalmology, Faculty of Medicine, Prince of Songkla University, Songkhla, Thailand. [4]Department of Material Science, Stanford University, Stanford, CA, USA. [5]Department of Electrical Engineering, Stanford University, Stanford, CA, USA. [6]Department of Physics, University of Strathclyde, Glasgow, Scotland, UK. [7]These authors contributed equally: Nicharee Monkongpitukkul, Andrew Shin. ✉e-mail: bhuckory@stanford.edu

However, with planar pixels having return electrode surrounding the active electrode in each pixel, such as in the PRIMA array (Fig. 1A, B), the distance between the active and return electrodes determines the penetration depth of the electric field into the retina[11]. Therefore, the stimulation threshold rapidly increases with a decreasing pixel size, and the needed light intensity exceeds the optical and electrochemical safety limits with pixels smaller than 75 μm for human retina[12,13].

We developed three different approaches to overcome this limitation; a planar monopolar (MP) array, and two three-dimensional geometries: honeycomb (HC) and pillar (PIL) devices with pixels sizes ranging from 55 μm to 22 μm. MP devices contain central active electrodes in each pixel and a common return electrode along the edge of the implant (Fig. 1C, D). During full-field illumination, the electric field is oriented vertically near the implant surface, resulting in a stimulation threshold 30 times lower (0.059 mW/mm$^2$) than with bipolar pixels of 40 μm[12]. High contrast and confinement of the electric fields can still be achieved in this configuration by current steering, using active electrodes on non-illuminated pixels as transient returns[14]. This strategy allows scaling the pixel size down to 22 μm in rats[15]. in the 3-D honeycomb configuration, the return electrode is elevated on top of 30 μm high isolating wall around each pixel, with the active electrode at the bottom of the well. The vertical electric field inside such cavities matches the direction of bipolar cells that migrate into the wells[11,16]. This effect reduces the stimulation threshold and decouples it from the pixel size. The final 3-D configuration involves elevating the active electrodes on top of pillar-like structures to achieve closer proximity to the BC in the INL after implantation. This approach might be the most beneficial for AMD patients, where INL is separated from the subretinal implant by about 40 μm of the debris layer[17].

We have characterized the retinal migration into the voids in subretinal implants in rats[18,19] and demonstrated that this process did not cause cell loss nor negatively affect the retinal excitability throughout the lifetime of the animals[16,20]. Our PRIMA implants are designed to last throughout the patients' lifetime, and already have shown stability in the subretinal space for over 4 years[9].

In this study, we demonstrate a possibility of safely removing the subretinal implants in all three configurations, in case they are no longer needed. Furthermore, we demonstrate feasibility of implanting the next-generation device in the same location to provide higher visual acuity. This opens the door to upgrading the current patients having the first-generation PRIMA arrays with 100 μm pixels, to the next-generation implant having 22 μm pixels (Fig. 1). We demonstrate that such a procedure preserves the retinal tissue and retains the stimulation thresholds, while increasing the visual acuity to the level of the new implant.

## Results

Three different subretinal implants: planar, honeycomb (HC) and pillar (PIL) arrays (Fig. 2A–C) were fabricated and implanted into the subretinal space of RCS rats (6 months old, n = 33) for 6 weeks, to ensure completion of the migration process of the inner nuclear layer (INL)[16]. Reconstructed confocal acquisitions of whole-mounted retina-implant-sclera complex were used to visualize cross-sectional views of 4′,6-diamidino-2-phenylindole (DAPI) labeled nuclei and their relative position. With planar devices, the whole INL was resting on top of the implant, with the outermost layer of cells in contact with the pixels (Fig. 2D). The honeycomb walls (Fig. 2E) and the pillars (Fig. 2F) maintained their mechanical integrity, with all structures intact after the implantation surgery, 6 weeks in vivo, and the following tissue preparations including its staining and imaging procedures. Retinal cells migrated and filled the HC cavities and the spaces between pillars, indicating strong physical interaction with the implants. Uniform integration with all geometries can be observed throughout the implant (Fig. 2G–I).

### Surgical explantation

To explore the feasibility of extracting the subretinal prosthesis, we developed a new surgical procedure. After the primary implantation, conjunctiva is sutured and the transscleral, transretinal cut underneath heals over 6 weeks. To avoid creating a second 1.5 mm cut in the 6 mm diameter eye, the incision for explantation was performed at the location of the healed scar of the primary surgery (see Supplemental Video 1). The retina was detached by gentle injection of balanced salt solution (BSS), followed by injection of viscoelastic gel to keep the subretinal space open. While visualizing the implant through the cornea, a 30-gauge blunt canula was inserted into the subretinal space next to the implant. A gentle stream of BSS was maintained until the retina had been detached from the edge of the implant. The canula was progressively moved between the retina and the implant until a complete detachment was observed. A second canula was inserted to coat the implant with viscoelastic gel as a form of lubrication to allow removing the implant smoothly. A 27-gauge Grieshaber surgical grasping forceps (Alcon) were inserted into the subretinal space to grab the edge of the implant and slowly pull it out, making sure that the retina remained undisturbed. The subretinal space was rinsed with BSS and the conjunctiva sutured. The surgeries for the three different implant geometries were comparable. The planar implants required less BSS to detach from the retina and were removed smoothly. HC implants took longer to detach but eventually, they could be dragged out of the subretinal space as smoothly as planar implants. PIL devices detached smoothly from the retina but required a continuous application of viscoelastic gel to separate retina from the top of the pillars, to avoid the retina getting stuck and damaged while dragging the device out. All implants remained undamaged and with preserved 3-D structures (Fig. 3A–F), and the explantation surgery did not leave any prosthetic pieces and materials in the subretinal space.

### Retinal integrity post device explantation

Removal of prosthetic devices that have integrated with biological tissue poses the risk of damaging the tissue and potentially aggravating the disease condition. To ensure that parts of the INL were not damaged and to assess the amount of biological tissue left on the

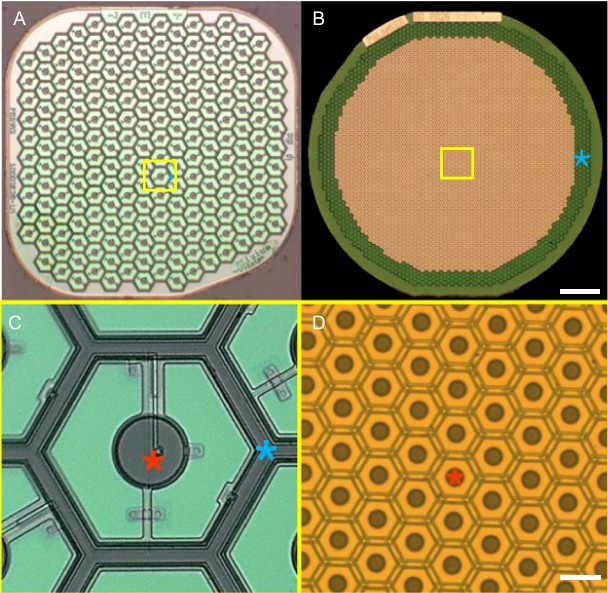

**Fig. 1 | Two generations of subretinal photovoltaic implants. A** A 1.5 mm wide PRIMA array with 100 μm pixels. **B** Monopolar flat array with 22 μm pixels. Scale bar 200 μm. **C, D** Higher magnification of the implants (yellow box areas). Scale bar 20 μm. Red asterisks—active electrodes, Blue asterisks—return electrodes in bipolar pixels of PRIMA, and a common ring return electrode of the monopolar array.

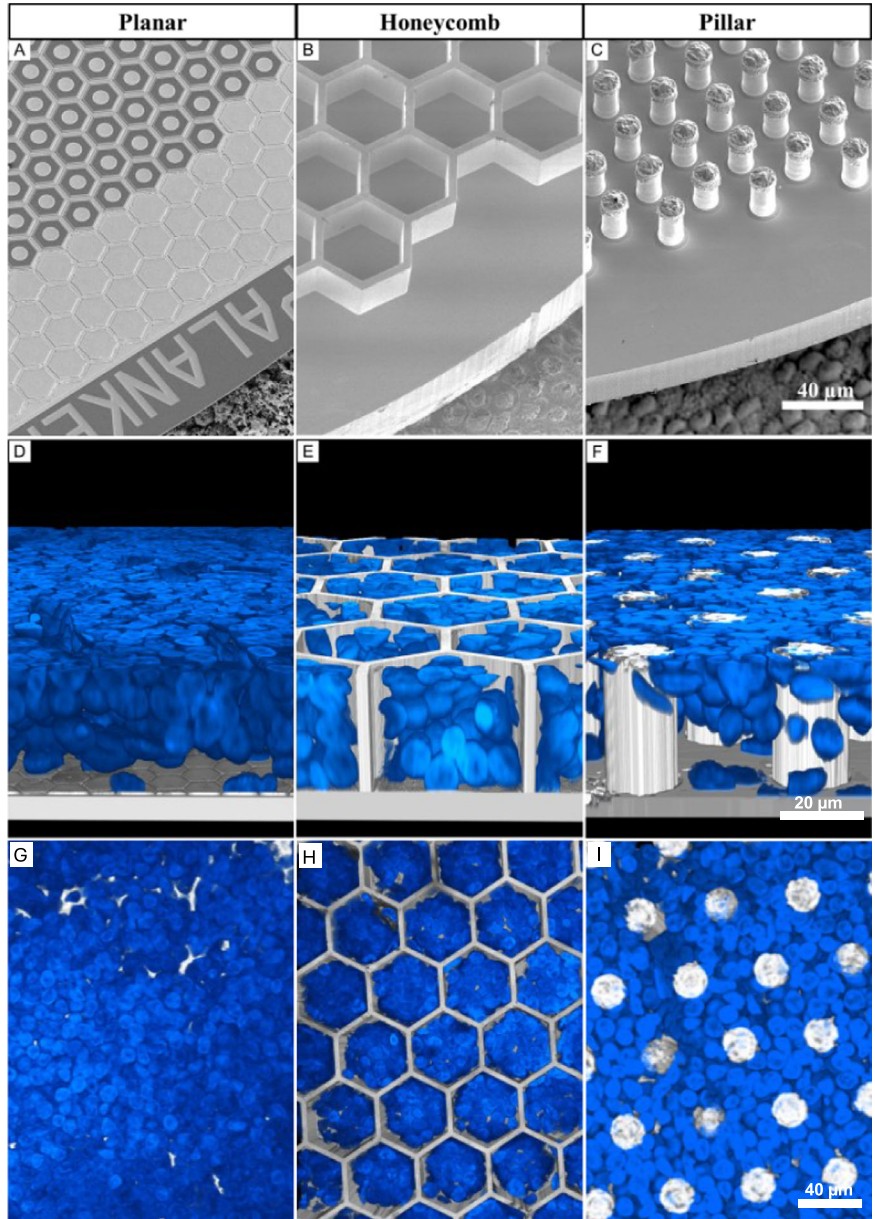

**Fig. 2 | Subretinal arrays of three geometries.** Scanning electron microscopy of **A** planar monopolar array, **B** 25 μm tall honeycomb structures fabricated in silicon and sputtered with titanium, **C** implants with electroplated 35 μm tall gold pillars sputter-coated with titanium. All implants are imaged on top of the porcine RPE for scale. This fabrication process has been established and repeated independently five times with similar outcomes for each type of implant. **D**–**F** Rendered confocal images of the whole mount retina on top of the implant, showing the inner nuclear layer (INL) up to the top of the 3-D array 6 weeks after implantation, indicating the level of retinal integration with each implant. Blue is the DAPI staining of cell nuclei. **D** Planar implant with the full INL on top ($n = 15$), **E** honeycombs with part of the INL cells within the wells ($n = 17$), **F** pillar array with the INL cells that migrated between the electroplated structures ($n = 6$). **G**–**I** Unrendered confocal images showing the top-down z-stack view of DAPI nuclei across the implant 25 μm above the base. $N$ numbers represent individual experiments with biological replicates, all yielding similar results.

devices after explantation, the implants were fixed in 4% paraformaldehyde (PFA) overnight and processed for scanning electron microscopy (SEM) imaging. Figure 3 demonstrates residual biological material on surface of the implants of all three geometries compared to pristine pre-implanted devices (Fig. 2A–C). Cell-shaped structures with rounded cell bodies and extending dendrite-like structures were observed in the higher magnification SEM images (red arrows; Fig. 3D–F). Acellular residues (yellow arrows) were also observed on the planar surface, inside the HC wells and between the pillars. Presence of cells on the extracted implants was confirmed by confocal imaging with DAPI nuclei staining (Fig. 3G–I). Due to their structural resemblance to immune cells, IBA1 staining was used to identify the cell type. Most of the cell nuclei co-localized with IBA1 (green; Fig. 3J–L), indicating that they were microglial cells rather than being from neuronal origin. These cells had attached to- and stayed on the implants after retinal detachment. Moreover, there was no apparent difference in the distribution of cells remaining on the implants for all three geometries.

The retinal integrity and recovery after removal of the prosthetic devices was monitored in vivo by funduscopic examination and by optical coherence tomography (OCT) imaging. The position of retinal blood vessels in the fundus relative to the original location of the implants was used to orientate and monitor the correct area. Six weeks after implantation, the subretinal arrays were visible through a clear

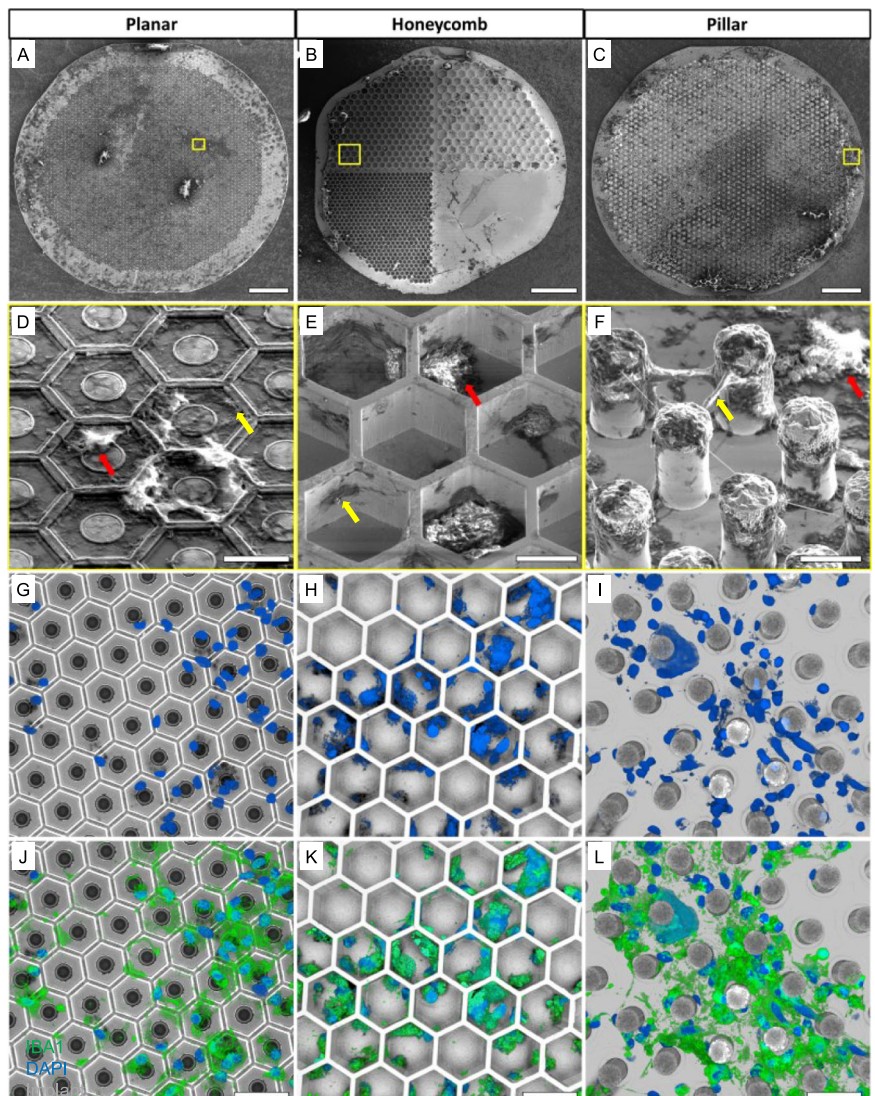

**Fig. 3 | Extracted devices.** Scanning electron micrograph (SEM) of **A** planar (*n* = 4), **B** honeycomb (*n* = 4), and **C** pillar (*n* = 4) devices extracted after 6 weeks in the subretinal space, indicating the distribution of biological remnants on the implants after surgery. Scale bar 100 μm. Magnified images **D**–**F** of the areas indicated by the yellow boxes show apparent acellular organic matter (yellow arrows) and cell-like structures (red arrows) on all three devices. Scale bar 15 μm. **G**–**I** Confocal microscopy of the implants (gray) and DAPI staining (blue) confirmed the presence of cell nuclei. **J**–**L** Most cell nuclei (blue) co-localized with IBA1 (green), indicating microglial cells. Scale bar 40 μm. *N* numbers represent individual experiments with biological replicates, all yielding similar results; all extracted implants (*n* = 4 per group) from **A**–**C** went through immunohistochemistry and confocal imaging (**G**–**L**), yielding similar results.

retina (Fig. 4A, G, M). OCT images confirmed the appearance of the inner retina above the implants. The INL was in close proximity to the surface of the planar implant (Fig. 4D), and with the 3-D implants, only part of the INL was visible above the implants, indicating complete migration of the INL into the cavities (Fig. 4J, P). After extraction of the implants, the retina had dark areas indicating the presence of blood or subretinal debris (Fig. 4B, H, N). The cross-sectional views of OCT confirmed the presence of subretinal materials (yellow arrows) preventing full reattachment of the retina to the retinal pigment epithelium (RPE) layer (Fig. 4E, K, Q). After six weeks of recovery, the retina appeared clear funduscopically (Fig. 4C, I, O), similar to the non-implanted RCS rat retina. The inner retinal layers were well preserved in all three geometries, with no significant difference in the INL thickness pre-extraction and 6 weeks after recovery from explantation (two-way ANOVA; Bonferroni post-hoc; *p* = 0.8). However, a thick hyper-reflective layer (red arrows) was present in the subretinal space, and occasionally, a dark 'pocket' was observed where the implant used to be (double white asterisks). The hyper-reflective/debris layer

appeared post-extraction after explantation with all three geometries and stabilized at similar subretinal thickness in all groups (Fig. 4T). On the day of extraction, both the INL thickness and debris layer thickness peaked, potentially from subretinal blood, fluid and retinal swelling from the surgery[21].

Histological analysis of the recovered retina six weeks after the explantation showed preserved INL cells (Fig. 5B–D) when compared to an age-matched non-implanted RCS retina (Fig. 5A). However, after extraction of 3-D implants (Fig. 5C, D), the INL appears to be less organized due to the induced migration into the implants and their removal. The hyper-reflective membrane on OCT (Fig. 4; red arrows) appeared as mostly acellular material in the subretinal space (Fig. 5; red arrows). The 'dark pockets' observed on OCT (Fig. 4; double white asterisk), remained unstained (Fig. 5B; black asterisk), suggesting that they were fluid filled pockets. Masson's trichrome staining (MTS), which differentially stains collagen, demonstrated that the collagen containing acellular layer (Fig. 5E–H; blue arrows), was only present in explanted locations and not in the control retina. Müller cell activation

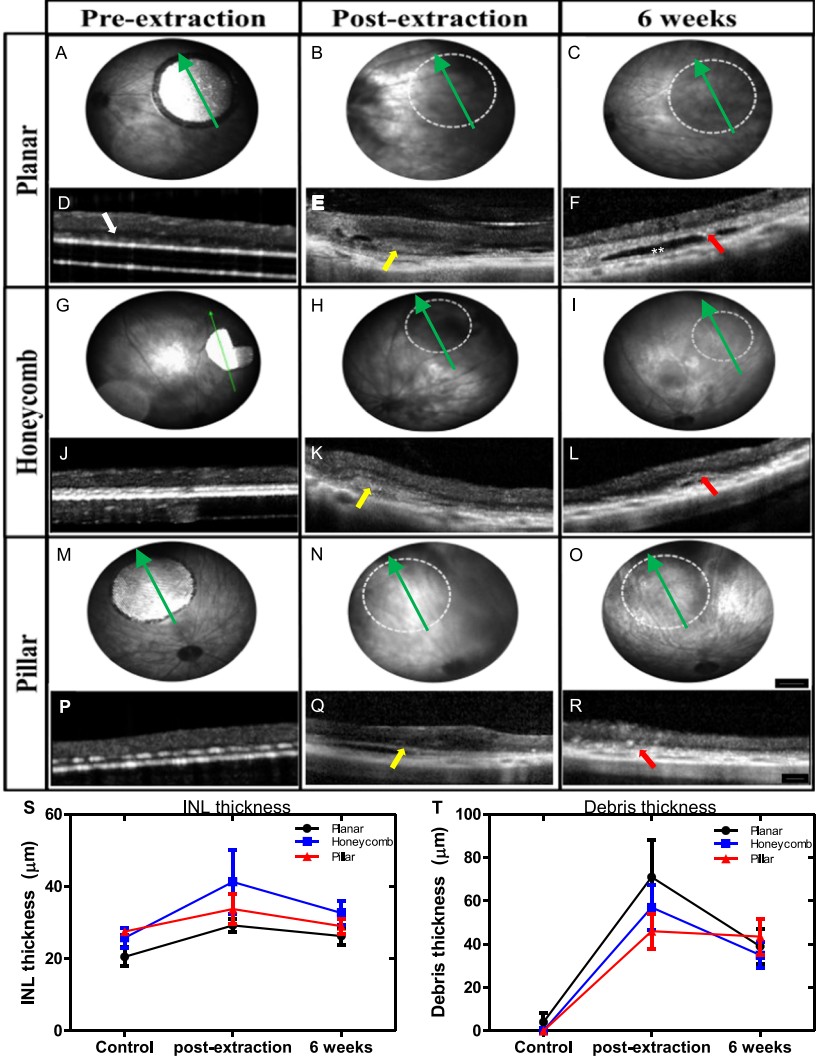

**Fig. 4 | In vivo post-explantation imaging.** Fundus images of the retina with the original implant (**A, G, M**), the day of extraction (**B, H, N**), and 6 weeks later (**C, I, O**) with planar (top row), honeycomb (middle row), and pillar (bottom row) implants. Location of the former implant (dotted white circle) cleared up during the 6 weeks recovery. Scale bar 500 μm. The cross-sectional view on OCT shows the close proximity of the INL with the planar implant (white arrow in **D**), and disappearance of the INL into the 3-D implants (**J, P**). Subretinal debris (yellow arrows) can be observed on the day of extraction (**E, K, Q**). After 6 weeks of recovery, retinal thickness is comparable in all three geometries (**F, L, R**). Red arrows indicate a thick hyper-reflective layer and, in some cases, a 'pocket' (double white asterisk). Green arrows indicated the OCT scan line. The INL thickness (**S**) and subretinal debris layer thickness (**T**) were quantified from biological replicated pre-extraction, on the day of the extraction surgery and after 6 weeks of recovery for the planar ($n = 4$), honeycomb ($n = 4$), and pillar ($n = 4$) groups. Two-way ANOVA; Bonferroni post-hoc; $p = 0.8$. Data are presented as mean values and error bars ± SEM. Scale bar 100 μm.

was labeled with glial fibrillary acidic protein (GFAP) as a marker of retinal response to injury. The retina having undergone explantation surgery displayed no significant (one-way ANOVA; Turkey's post hoc; $p = 0.05$) change in GFAP expression compared to the RCS control (Fig. 5I–L, supplementary Fig. 1I), suggesting the baseline activation of Müller cells is related mostly to the disease pathology rather than surgery. A glial membrane was observed (Fig. 5 yellow arrows) in the subretinal space locations where collagenous subretinal fibrosis was identified. However, it was also present in the RCS control. Müller cell staining by glutamine synthetase also showed similar morphology in RCS rat retina, but with varying levels of INL-subretinal glial seal (Fig. 5M–P). After 6 weeks of recovery and without the presence of a subretinal foreign body, the exacerbated infiltration of microglial cells observed on the extracted implants on the day of explantation (Fig. 3J–L) was no longer significantly present in the subretinal space when compared to control (supplementary Fig. 1A–D, J; one-way ANOVA; Turkey's post hoc; $p = 0.08$). After the degeneration of

photoreceptors, the sparsely present horizontal cells are no longer connected and appear disorganized (supplementary Fig. 1E). The implantation and extraction of subretinal implants did not further affect the horizontal cells and their dendrites (Supplementary Fig. 1F-H).

## Implant replacement and upgrade

The ability to safely remove implants from the subretinal space opens the possibility of replacing the implants with improved new generation devices. PRIMA with 100 μm pixels has been an important clinical proof of concept for photovoltaic subretinal prosthesis. Clinical studies with AMD patients demonstrated that prosthetic visual acuity matched the 100 μm pixel size of these implants[9,10]. Since animal studies with our next generation devices demonstrated much higher resolution, we assessed the feasibility of replacing the PRIMA implants with flat arrays of 22 μm pixels. Six weeks after the initial subretinal implantation in RCS rats, PRIMA implants were extracted following the extraction

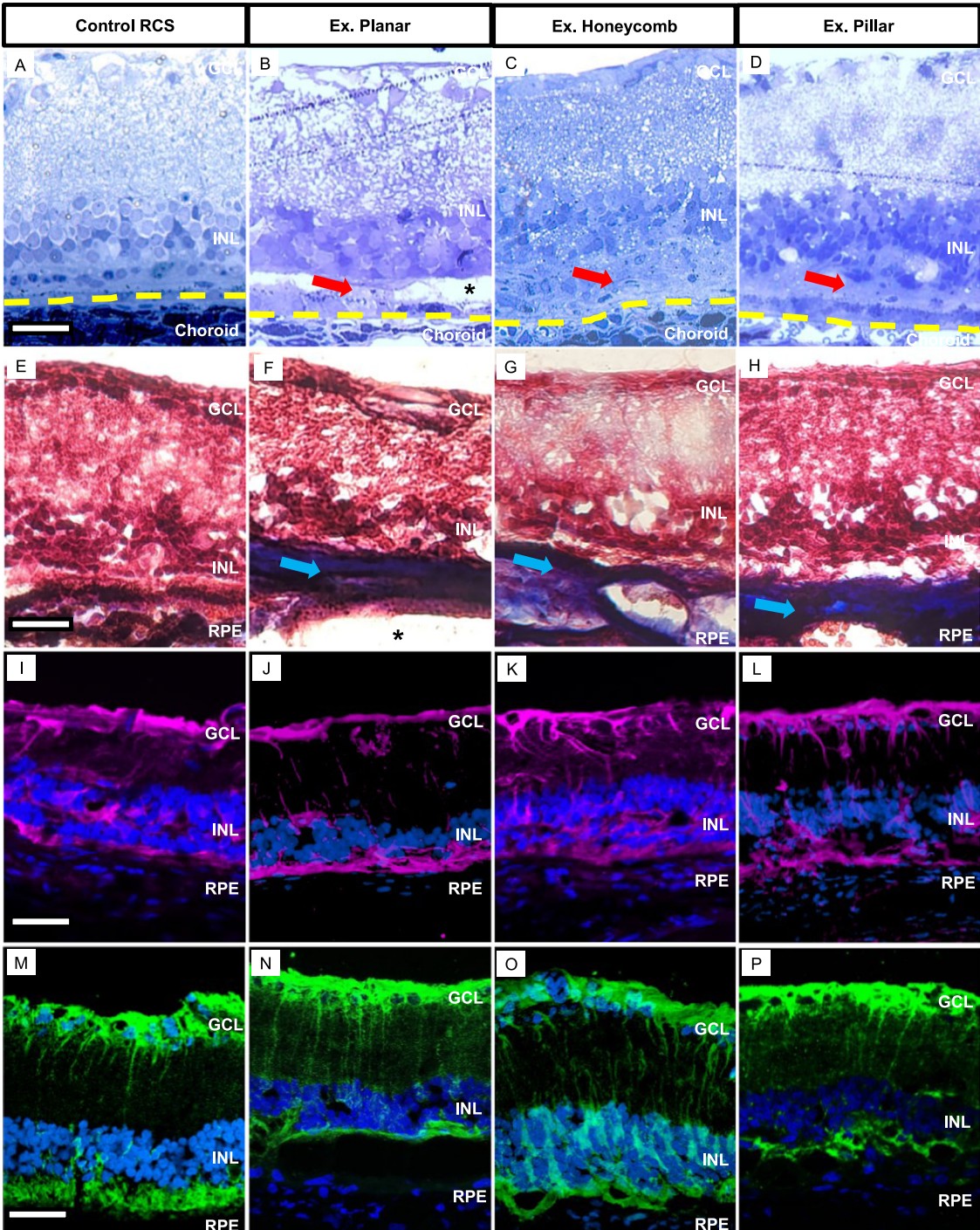

**Fig. 5 | Histological characterization post-explantation.** Toluidine blue-stained histological sections 6 weeks post-explantation demonstrate preservation of the INL with planar, honeycomb and pillar (**B**–**D**; n = 4 per group) devices, compared to non-implanted RCS control (**A**; n = 4). A thick acellular layer (red arrows) develops in the subretinal space after explantation. A pocket (*) where the implant used to be is visible in some sections. The yellow line demarks the RPE/choroid boundary. Scale bar is 50 μm. Masson's trichrome staining (MTS) labeled the acellular layer as collagen in blue (blue arrows). It was present after removal of all three devices (**F**–**H**; n = 4 per group), but not in the RCS control (**E**; n = 4). Scale bar is 70 μm. GFAP (magenta) immuno-labeling of the sections showed the Müller cell activation between an RCS control (**I**; n = 8) and the three device groups (**J**–**L**; n = 3 per group). Scale bar is 80 μm. Müller glial population (**M**–**P**; n = 4 per group) labeled with glutamine synthetase (green) show retracted cells and the appearance of a glial seal after photoreceptor degeneration in the RCS control. Similar Müller cells are observed after extraction of planar, honeycomb and pillar implants, with varying levels of glial seal. Scale bar is 80 μm. N numbers represent individual experiments with biological replicates, all yielding similar results. INL inner nuclear layer, GCL ganglion cell layer, RPE retinal pigment epithelium.

procedure of planar devices we described. After the primary implant was out, an MP device was coated with viscoelastic gel and inserted between the retina and RPE through the same 1.5 mm incision. While visualizing the implant through the cornea, it was carefully slid into the exact same location as the primary device, defining the implant placement based on location of the retinal blood vessels nearby (Fig. 6A–F). Fundus images shown in Fig. 6G, H confirm the correct placement of the arrays (yellow arrows). OCT images showed well preserved retinal layers both, pre-extraction and post-reimplantation, with no significant difference in the INL thickness (Fig. 6I; one-way

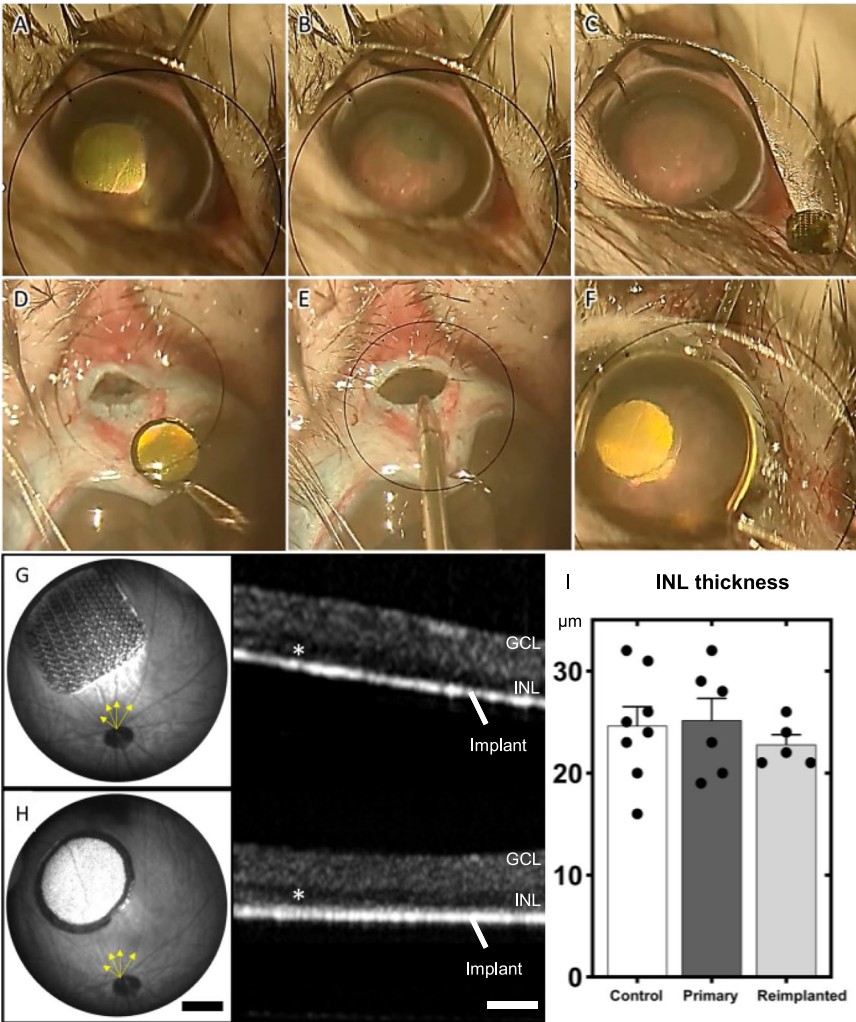

**Fig. 6 | Implant removal and upgrade.** The key surgical steps of the removal of a PRIMA 100 μm implant and replacement with a monopolar (MP) 22 μm implant, including: **A** retinal detachment from the PRIMA implant by BSS injection, **B** grabbing the implant and dragging it in the subretinal space, **C** extraction of the PRIMA implant through the 1.5 mm incision in the sclera, **D** coating the MP device with viscoelastic gel, **E** inserting the implant through the same incision, **F** placing the new device in the same location as the PRIMA. Implants are marked by dotted lines. Fundus (scale bar 500 μm) and OCT image (scale bar 100 μm) of a PRIMA primary implant after 6 weeks of implantation (**G**) and an MP device 6 weeks after replacing the PRIMA implant (**H**). Yellow arrows indicate blood vessels used to orientate the new device in the same location as the primary implant. **I** Quantification of the INL thickness from biological replicates of RCS control ($n = 8$), primary implanted retina ($n = 6$), and 6 weeks post-reimplantation ($n = 5$). One-way ANOVA; Turkey's post hoc; $p = 0.5$. Data are presented as mean values and error bars ± SEM. INL inner nuclear layer, GCL ganglion cell layer.

ANOVA; Turkey's post hoc; $p = 0.5$). Implant replacement immediately after extraction did not cause subretinal fibrosis or development of a subretinal pocket, as observed when the retina was allowed to recover for the same amount of time but without an implant (Fig. 4; red arrows).

While the implant replacement surgeries were successful and retained the inner retinal anatomy, retinal function, excitability and visual acuity remain the key measures of success for retinal prostheses. We measured the visually evoked potential (VEP) in response to full field illumination (Fig. 7A) and defined the stimulation threshold–the lowest light intensity that elicits a significant cortical response (Fig. 7B). VEPs were recorded via transcranial electrodes above the visual cortices, with NIR stimuli applied at 2 Hz with 10 ms pulse duration, while varying the irradiance. The upgrades were from the PRIMA 100 μm pixel arrays of 1.5 mm in width to the new generation arrays of the same width, having 22 μm monopolar (MP) pixels. Since the PRIMA and MP arrays have very different field confinement and stimulation thresholds, we compared the retinal excitability before and after the replacement surgery by comparing the responses to the same MP devices, implanted once or re-implanted. Reimplanted MP devices showed very similar responses at all light intensities and importantly, the stimulation threshold of both groups was 0.06 mW/mm² (Fig. 7E), suggesting that the retina retains its electrical excitability after the second surgery. Furthermore, we assessed the improvement in spatial resolution by measuring the grating acuity using alternating gratings projected on the reimplanted devices (Fig, 7B). The logarithmic fitting curve crossed the noise level at the 26 μm mark for the primary implantation of MP22 and 28 μm mark after reimplantation (Fig. 7F), closely matching the 28 μm natural limit of visual acuity in rats[19,20] and our previous results with primary MP22 implants[15]. This represents a significant improvement in visual acuity compared to the 87 μm pixel pitch of PRIMA devices ($d = 100$ μm cos 30° for hexagonal array).

## Discussion

Clinical trials with PRIMA implants having 100 μm pixels (Pixium Vision SA, Paris, France) demonstrated restoration of central vision in patients blinded by atrophic AMD, with highest resolution achieved to

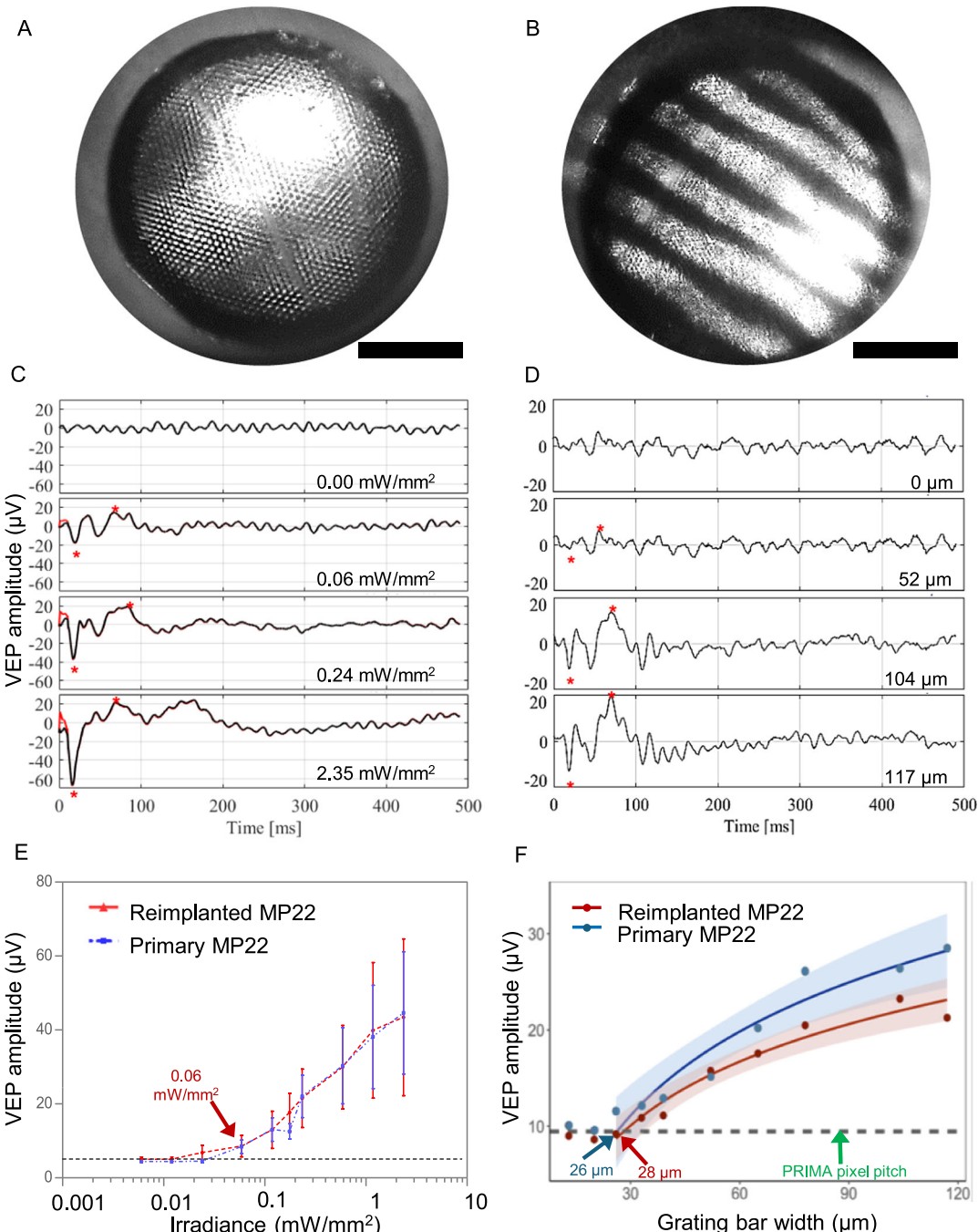

**Fig. 7 | Electrophysiological measurements.** Visually evoked potential (VEP) measurements in RCS rats ($n = 4$ per group) were performed by a full-field illumination (**A**) and alternating grating patterns (**B**) projected onto the reimplanted MP devices (scale bars 500 µm). **C** Representative waveforms for the full-field stimulation using 10 ms pulses at 2 Hz with varying light intensities. Above the threshold, the characteristic VEP peaks appear (red asterisks). **D** Example waveforms of a response to alternating gratings (red asterisk). **E** Peak-to-peak VEP amplitude as a function of irradiance for a primary implanted MP 22 µm (blue) and a reimplanted MP 22 µm device (red). VEP signal above noise (dotted black line) was observed at 0.06 mW/mm² in both measurements. **F** VEP amplitude as a function of the grating bar width for primary implanted and reimplanted MP 22 µm devices. Acuity limit defined as the point where the log fit crosses the noise level (26 and 28 µm, respectively). The error bars represent the standard error of mean (SEM). Each group had 7 biologically independent subjects. The black dash lines represent the mean noise level; red and blue lines are logarithmic fit and the bands around them represent the 95% confidence interval.

date by visual prostheses: up to 20/438, closely matching its pixel size limit of 20/420[9,10]. While very promising for the field and life changing for the patients, visual acuity should increase further to exceed the legal blindness threshold of 20/200, and ideally even beyond. Our next generation implants with pixel sizes down to 22 µm, have demonstrated feasibility of achieving this goal in rats as well as in computational models of human retina[11,15,16,20,22]. Therefore, it is necessary to

determine the feasibility and safety of the possible explantation of subretinal prosthesis and its exchange with a newer version. It is especially challenging with 3-D configurations of the devices that have shown extensive cell migration of the retina into the implants (Fig. 2)[16,22]. Planar PRIMA implants were designed to last the lifetime of the patients and have shown good stability for over 7 years so far[9]. Similar long-term subretinal stability is expected with 3-D arrays[20].

However, in the event of an implant malfunction, elective removal, or if a new generation device with higher visual acuity limit becomes available, explantation or replacement of these subretinal implants may be necessary. Our study has shown the feasibility of safely removing devices of all three configurations. We also demonstrated reimplantation of the next-generation arrays instead of the first-generation flat PRIMA implants and validated the subsequent improvement in spatial resolution.

To avoid unnecessary damage to retinal vasculature and peripheral cells, the retina was incised and re-detached at the primary incision scar location. Planar implants remained stable in the subretinal space (Figs. 2D and 4A), held in place by the forces keeping the retina attached: RPE fluid transport to choroid and by ocular pressure. No resistance was encountered when injecting BSS between the retina and the implant and the device could be readily removed. HC implants were completely attached to the retina and required a slow, steady stream of BSS, while progressively moving the irrigation canula between the retina and top of HC walls. Once detached, its removal was comparable to planar implants, sliding smoothly out of the subretinal space. PIL implants, on the other hand, required more gentle manipulations. The PILs are fragile structures, and care must be taken when inserting the cannula between the retina and the pillar tops. Furthermore, even after detachment of the retina from the implant, the PILs had a tendency to stick to the retina and required constant injection of a viscoelastic bolus. The rat lens occupies the majority of the posterior chamber, leaving limited space (approximately 1 mm; supplementary Fig. 3) for retinal detachment, insertion of tools and manipulation. In larger animals and humans, the significantly larger posterior chambers will allow bigger blebs to be made during retinal detachment to facilitate subretinal manipulations. Additionally, lens removal and vitrectomy in the larger eyes could further help the surgeries. For PIL implant removal, a specialized tool that grabs the edge of the implant and has a protective cover separating the top of the PILs from the retina could further facilitate the removal process and reduce the risk of retinal injury.

The promising results with the next generation implants in terms of safety and resolution[15,20,22] suggest that they could reach clinical trials soon. If successful as a commercial product indicated for restoring sight, patients with the first-generation PRIMA devices should also be able to benefit from this advancement. The ability to upgrade from PRIMA 100 μm devices to new implants with 22 μm pixels could drastically improve the prosthetic visual acuity of these patients from the current 20/438 limit[10] up to 20/80[15,16]. Tiling such implants to the full size the scotoma should increase the field of view, while the digital zoom on the external camera should allow these patients to get close to acuity of 20/20. Our results demonstrate that the retina retains its structure and its electrical excitability after the upgrade surgery from PRIMA to a planar MP device, and that visual acuity with 22 μm pixels reaches the 28 μm limit of visual resolution in rats. Photovoltaic HC and PIL devices are in the design and fabrication phases, and tests of the upgrade from PRIMA to HC and PIL will be conducted once these devices have been fabricated.

While the retinal morphology was preserved after explantation of the devices, the eyes developed subretinal fibrosis at the location of former implant (supplementary Fig. 2). Subretinal fibrosis was not observed when a secondary implant was immediately placed at the same location in the subretinal space (Fig. 6G, H). The known contributors to subretinal fibrosis include Müller, microglial, infiltrating immune cells, complement activation, pericytes and RPE cells[23–26]. If Müller and microglial cells were significant contributors after the implant extraction, reimplantation of a new device would still result in a hyper-reflective seal forming between the implant and the retina. Since it was not observed, it seems likely that the mediators were infiltrating from the RPE/choroid side and separation by the implant prevents their migration towards the retina. Due to the implant opacity, OCT cannot visualize the space behind the device, thus further histological studies will be required to assess whether the fibrotic membrane observed is still forming behind the implant after the upgrade. Subretinal fibrosis could diminish the chances of success in later attempts to restore sight by other means. If the mediators of subretinal fibrosis after prosthesis explantation can be identified, subretinal drug dosing at the time of the surgery might help prevent the formation of these fibrotic membranes.

In conclusion, our study demonstrated the feasibility of safely removing subretinal implants of planar and 3-D configurations and replacing them with a new generation device. After re-implantation, retinal excitability remained at the original level, while resolution increased up to the natural limit of visual acuity in rats. The developed surgical techniques can be adapted and should be easier to perform in much larger human eyes. The capability of upgrading the retinal protheses may alleviate the potential concern that implantation of the early versions of the device may preclude patients ability to use the next-generation implants.

## Methods
### Planar implants
Planar implants of 2 configurations were used in this study: PRIMA implants with photovoltaic pixels of 100 μm containing local return electrodes around each pixel, and monopolar (MP) devices of 22 μm pixels containing global returns around the edge of the implant (Fig. 1). The 1.5 mm PRIMA implants were fabricated by and obtained from Pixium Vision[9,10]. The 1.5 mm monopolar devices were fabricated at Stanford as previously described[27]. Briefly, implants are optimized for 880 nm wavelength: with 30 μm thickness and metallization on the back surface of the device, approximately 90% of light is absorbed in Si. These photodiode arrays consist of 3508 hexagonal pixels of 22 μm in width. About 20% of the pixels at the implant's periphery are coated with a common return. The fabrication process involves creating a vertical p–n junction, establishing ohmic contact to the electrodes, applying anti-reflection coatings and metal electrodes, adding a high-capacitance coating (SIROF) to the electrodes, and releasing the device[27]. Subsequently, the sidewalls and back side of the devices are coated with titanium for protection from erosion.

### Honeycomb-shaped implants
The inner nuclear layer (INL) thickness in RCS rats ranges from 40 to 50 μm and consists of 4–5 layers of nuclei. We have previously demonstrated that 25 μm deep wells of the honeycombs allow bipolar cells to migrate into the cavities while keeping amacrine cells (AC) out[16]. For anatomical investigations, passive honeycomb implants were constructed from crystalline silicon using a two-layer mask technique for generating deep silicon etching patterns (Fig. 2B), as previously outlined[11].

Briefly, a hexamethyldisilazane (HMDS; Wacker Chemie AG, Burghausen, Germany) primed wafer underwent a spin-coating process with 2 μm of negative photoresist (AZ5214-IR; Integrated Micro Materials, USA). This layer was exposed to UV light through a patterned photomask (Applied Materials, USA), leading to the creation of 25 μm deep cavities using a Bosch etch process (Inductively Coupled Plasma ICP Etching Systems; Plasma-Therm, USA) in the unprotected regions. After removing the honeycomb-defining resist, a second photoresist layer (7.5% SPR 220-7, 68% MEK, and 24.5% PGMEA; Kayaku Advanced Materials, USA) was spray-coated onto the wafer to a thickness of 30 μm. This layer was exposed to define the releasing trenches around the 1 mm wide arrays, also utilizing a Bosch process.

Subsequently, the wafer received a protective spray-coating of 60 μm thick photoresist for backside grinding from 500 to 50 μm, starting from the base of the honeycombs. The remaining excess silicon was etched in XeF2 gas (Xactix; SPTS Tech, USA) to complete the release of the implants. Each implant comprised four quadrants,

featuring hexagonal honeycomb patterns of 40, 30, and 22 μm in width with walls of 25 μm height, having thicknesses of 4, 3, and 2 μm, respectively. The fourth quadrant served as a flat control, as depicted in Fig. 3B. To prevent the dissolution of thermally oxidized silicon (300 nm) in-vivo, arrays were sputter-coated with 200 nm of titanium.

## Pillar implants

Pillar electrodes were electroplated (NB Semiplate AU 100TH; NB Technologies, Bremen, Germany), as previously described[22]. Briefly, patterns for the active and return electrode structures were created using a Ti:Au layer (50 nm:200 nm; Kurt J. Lesker, USA) on blank 4-inch silicon wafers (p-doped). The interconnected active and return electrode structures enabled electroplating the 3D devices across the entire wafer simultaneously. Each pixel's disk electrode was electroplated into a pillar with a 16.5 μm diameter for 40 μm pixels. Applying a constant current density of 1 mA/cm² to the patterned wafers resulted in a plating rate of 3 μm per hour, achieving the desired pillar height in gold. The top surface of the electroplated structures was subsequently sputter-coated with Ti:SIROF (40 nm:436 nm; EIC Lab, MA, USA) to provide a high-capacitance material for the electro-neural interface. For anatomical integration studies, the whole implant was sputtered with 200 nm Ti before implantation into the subretinal space.

## Animals and surgical procedures

All experimental protocols received approval from the Administrative Panel on Laboratory Animal Care (APLAC) at Stanford and were executed following institutional guidelines. The procedures adhered to the Statement for the Use of Animals in Ophthalmic and Vision Research, as outlined by the Association for Research in Vision and Ophthalmology (ARVO). Animal care and implantation procedures were conducted in accordance with previously documented methods[6,28]. Royal College of Surgeons (RCS-p+/LavRrrc; RRID:RRRC_00315, breeders purchased from RRRC, Missouri University; colony maintained at the Stanford Animal Facility) rats with a genetic mutation in MERTK gene resulting in complete photoreceptor degeneration between 4 and 6 months of age, were housed at the Stanford animal facility.

For anatomical studies, a total of $N = 38$ animals were implanted subretinally with different types of arrays (15 planar, 17 honeycomb, and 6 pillar implants), and the devices were explanted 6 weeks later. Extraction studies, OCT and immunohistochemistry had 4 animals per group. Reimplantation studies had 8 controls, 6 primary implant, and 5 reimplanted implants were implanted with a PRIMA 100 μm chip for 6 weeks and the array was replaced with a 22 μm planar implant and monitored for up to 6 months. VEP studies had 4 animals per group. Experiments maintained equal numbers of male and female rats.

For the surgical procedures, animals were anesthetized with a combination of ketamine (75 mg/kg; VetOne, USA) and xylazine (5 mg/kg; Vetone, USA) administered intraperitoneally. A 1.5 mm incision was made through the sclera and choroid approximately 1 mm posterior to the limbus. Retinal detachment was induced with a saline solution injection (BSS, Alcon, USA), and the implant was placed into the subretinal space and positioned away from the incision site. The conjunctiva was sutured using nylon 10-0 (Ethicon, J&J, USA), and topical antibiotic (bacitracin/polymyxin B; Bausch and Lomb, USA) was applied to the eye postoperatively.

## Imaging

**In-vivo OCT.** Surgical success and retinal reattachment on top of the array was confirmed through Optical Coherence Tomography (OCT) (HEYEX v.1.12.40; HRA2-Spectralis; Heidelberg Engineering, Heidelberg, Germany). For ease of imaging in rat eyes, the cornea was covered with viscoelastic gel (Viscoat, Alcon, USA) and a coverslip (VWR, USA) was used to cancel the corneal curvature and its optical power. OCT monitoring was conducted regularly to observe the recovery of

the retina after initial surgery, before extraction of the implants and weekly for 6 weeks post-surgery.

**Scanning electron microscopy (SEM).** Implants of all three configurations were imaged before implantation and after extraction using a Zeiss Sigma SEM with Schottky field emission (FE) source and GEMINI electron optical column (Ziess, Oberkochen, Germany). To assess the material left on the devices after surgical explantation, the devices were fixed in 4% PFA (VWR, USA) overnight and then processed for SEM imaging. The samples are put on a sample holder with carbon tape so that the light samples are not blown away during the SEM chamber pump process (FEI Helios NanoLab 600i DualBeam SEM/FIB and Thermo Fisher Scientific Apreo S LoVac Scanning Electron Microscope). The sample holder with implants is mounted in the SEM chamber and pumped down below 9E-5 mbar. Once a vacuum is established, E-beam with 2 kV, 43pA is applied on the sample for imaging. Usually, lower beam power is applied for better imaging on the interface level (so e-beam does not penetrate too deep and image the inner layers).

**Confocal fluorescence imaging of whole-mount retina.** Whole-mount preparations of the retina integrated with the implants were stained with 4′,6-diamidino-2-phenylindole (DAPI; Thermo Fisher Scientific, Rockford, IL). After extraction surgery, the implants were fixed in 4% PFA and immune labeled with 1:400 rabbit raised IBA1 antibody (WAKO, Japan) at room temperature for 24 hours. To label the implants, 1:400 donkey raised rabbit secondary antibody conjugated with Alexa Fluor 488 (Thermo Fisher Scientific, Rockford, IL) and DAPI were incubated for 24 h at room temperature. Three-dimensional imaging of retinal whole mounts was conducted using a Zeiss LSM 880 Confocal Inverted Microscope with Zeiss ZEN Black software (Zeiss, Germany). The identification of implant surfaces was achieved by reflecting a 514 nm laser and using a neutral-density beam splitter that allowed 80% transmission and 20% reflection. Images were acquired through the entire inner nuclear layer (INL) thickness using Z-stack, 10 μm above the INL and below the base of the devices. Stacks were obtained at the center of each implant, with a 40× oil-immersion objective and an acquisition area of 225 × 225 μm at 500 nm z-steps. The Zeiss z-stack correction module was utilized to compensate for lower light intensity within the wells of the implants.

Confocal fluorescence datasets were processed in ImageJ. To address brightness variations at different Z positions within the wells and above the implant, contrast maximization was initially applied to individual XY planes, ensuring 0.3% channel saturation. XY planes underwent de-speckling through the median filter, and background suppression was performed using the rolling-ball algorithm[29]. Subsequent cascades of gamma adjustments and min-max corrections were applied to further mitigate the background, relative to the noise level. Gaussian blurring was specifically implemented for nucleus staining channels to smooth brightness variations within individual cells. The reconstruction of implants involved projecting the implant reflection toward the bottom of the implant[30].

**Histological preparations.** After implant extraction, the eyes were allowed to recover for 6 weeks, then enucleated and fixed in a 1.25% glutaraldehyde solution (Sigma Aldrich, USA) for 24 h at room temperature. Subsequently, they underwent post-fixation in osmium tetroxide (Sigma Aldrich, USA) for 2 h at room temperature and dehydration through a series of graded alcohol and propylene oxide (Sigma Aldrich, USA). After overnight infiltration in epoxy (without DMP-30; SPI Supplies, USA) at room temperature with Electron Microscopy Sciences' Araldite-EMbed (RT13940, Mollenhauer's kit), the samples underwent a 36-h curing process in an oven at 70 °C. Epoxy blocks were carefully trimmed until the extracted implant area became visible. The resulting 700 nm thick sections, cut using a

Reichart UltracutE, were stained with 0.5% toluidine blue (S25612,Thermo Fisher, USA) for light microscopy imaging.

**Immunohistochemistry in frozen sections.** The eyes were enucleated after euthanasia and fixed in 4% (PFA) overnight. After removal of the cornea and lens, the explanted area was selected under a stereoscope and embedded in optimal cutting temperature compound. Once frozen, the block was sectioned at a thickness of 12 μm. The sections underwent permeabilization with triton-x and were incubated overnight at 4° with (1) 1:500 of goat raised glial fibrillary acidic protein primary antibody (GFAP; SC-6170; Santa Cruz Biotechnologies, Santa Cruz, CA), (2) 1:100 Mouse anti-glutamine synthetase (GS; NBP2-43646; Novus, CA), (3) 1:400 rabbit raised IBA1 antibody (WAKO, Japan), 1:100 mouse anti calbindin antibody (Swant; CB300; CA). Subsequently, the sections were incubated for 2 h with (1) 1:400 donkey raised anti goat, Alexa Fluor (AF) 594 conjugated secondary antibody (A-11058; Thermo Fisher Scientific, Rockford, IL), (2) 1:500 donkey anti-rabbit AF488 (A-21206; Thermo Fisher Scientific, Rockford, IL), (3) donkey anti mouse CY3 conjugated secondary antibody (715-165-150; Jackson labs, USA), along with DAPI. Confocal microscopy (LSM880; Zeiss, Jena, Germany) and Airyscan (LSM880; Zeiss) were used for imaging.

**Trichrome staining.** A trichrome stain kit (ab150686; Abcam, Boston, MA) was optimized for use on frozen sections. Briefly, the embedding medium was washed with PBS and slides were incubated in preheated Bouin's Fluid for 60 min and cooled for 10 min, followed by 3 rinses in water. Slides were then incubated in Weigert's Iron Hematoxylin for 5 min, rinsed in water, incubated in Biebrich Scarlet/Acid Fuchsin solution for a further 15 min, rinsed in water, differentiated in phosphomolybdic/phosphotungstic acid solution for 10–15 min, incubated in Aniline Blue solution for 5–10 min and rinsed 3 times in water. Finally, slides were incubated in acetic acid solution for 3–5 min, dehydrated, cleared and mounted for light microscopy imaging.

### Electrophysiology

To assess the visually evoked potentials (VEP) elicited in animals after PRIMA 100 μm implants were upgraded to monopolar 22 μm implants, each animal was implanted with three transcranial electrodes. One electrode was placed in the skull above each hemisphere's visual cortex (V1; 4 mm lateral from midline, 6 mm caudal to bregma), and one reference electrode above the somatosensory cortex (2 mm right of midline and 2 mm anterior to bregma).

Animals were lightly anesthetized and their pupils were dilated. An artificial tear gel and a cover slip were used to cancel the cornea's optical power. A customized projection system was used for visualizing and projecting the stimulation patterns onto the subretinal implant. It comprised a near-infrared laser with a wavelength of 880 nm (MF_880 nm_400 μm, DILAS, Tucson, AZ), collimating optics, and a digital micromirror display (DMD; DLP Light Commander; LOGIC PD, Carlsbad, CA) for generating optical patterns, coupled with a slit lamp (Zeiss SL-120; Carl Zeiss, Thornwood, NY) and a CCD camera (acA1300-60gmNIR; Basler, Ahrensburg, Germany).

Near-infrared (NIR) light with a pulse duration of 10 ms at 2 Hz, and irradiance ranging from 0.002 to 4.7 mW/mm² was used to determine the stimulation threshold of the first time implanted eyes with monopolar 22 μm arrays and compared to that with the same implants after re-implantation. The irradiance at the retina was calculated and adjusted based on the ratio between the sizes of projected pattern on the retina and on the cornea. VEPs were recorded using the Espion E3 system (Diagnosys LLC, Lowell, MA) at a sampling rate of 2 kHz and averaged over 500 trials. The stimulation threshold was defined as the VEP amplitude exceeding the noise above the 95% confidence interval[15,16,27].

Visual acuity was measured using projection of alternating grating patterns with various bar widths and determining the narrowest grating bars that elicited VEP signal. NIR light at 2.4 mW/mm², 4 ms-long pulses, at a carrier frequency of 64 Hz and grating switching cycle of 1 Hz (500 ms per image) were used as the projection parameters. Grating bar widths ranged from 13 μm to 117 μm on the retina. The noise baseline was determined by projecting static gratings with a 120 μm bar width, with the other stimulus parameters unchanged.

VEP data analysis was conducted using a custom code developed in MATLAB and Python[31]. The high-frequency noise corresponding to the pulsed stimuli (64 Hz carrier frequency) was filtered out using a spectrum reconstruction algorithm. The intersection of a logarithmic fit of the data points above the noise level with the baseline noise determined the grating visual acuity limit[15,16].

### Statistics and reproducibility

To determine the confidence interval of the acuity limits in rats, we first fit the VEP amplitudes to a linear function of grating width on a logarithmic scale (Fig. 7), with a curve_fit() function from optimization module of Scipy (version 1.7.1) in Python 3.8.8, which reports the covariance matrix of the fit parameters. The variance of the VEP fit value was then calculated as a function of the grating width. At the intersection between the VEP fit line and the noise level, defined as the nominal acuity limit, we used the statistical delta method to find the standard deviation of the acuity limit from those of the fit line and the noise level. By the normal approximation, the 95% confidence interval of the acuity limit was then determined as 1.92 times its standard deviation on each side of the intersection. Other statistical analyses were conducted using one-way analysis of variance (ANOVA) to evaluate differences among groups. Post hoc comparisons were performed using Tukey's Honestly Significant Difference test to identify specific group differences where applicable. A significance level of $p < 0.05$ was considered statistically significant. All statistical analyses were performed using GraphPad Prism 10 or R version 4.x. All experiments were replicated multiple times (n number provided as biological replicates in the Methods animal section) to ensure reproducibility.

### Ethics

Every experiment involving animals, human participants, or clinical samples have been carried out following a protocol approved by an ethical commission.

### Reporting summary

Further information on research design is available in the Nature Portfolio Reporting Summary linked to this article.

## Data availability

All data supporting the findings of this study are available within the article and its supplementary files. Any additional requests for information can be directed to, and will be fulfilled by, the corresponding authors. Source data are provided with this paper. The data that support the findings of this study are also available on the Zenodo database. Source data are provided with this paper.

## Code availability

Codes for VEP data processing are available from Zenodo and for implant reconstitution after confocal imaging at https://doi.org/10.5281/zenodo.14728769 under license Creative Commons Attribution 4.0 International and are available from the corresponding author upon request.

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

## Acknowledgements

The authors would like to thank Bing-Yi Wang for her help with VEP training during the pilot stages of the study. Studies were supported by the National Institutes of Health (Grants R01-EY-035227; D.P., and P30-EY-026877; D.P.), the Department of Defense (Grant W81XWH-22-1-0933; D.P.), AFOSR (Grant FA9550-19-1-0402; D.P.), Wu Tsai Institute of Neurosciences at Stanford; D.P., and unrestricted grant from the Research to Prevent Blindness; D.P. Photovoltaic arrays were fabricated at the Stanford Nano Shared Facilities (SNSF) and Stanford Nanofabrication Facility (SNF), which are supported by the National Science Foundation award ECCS1542152; Stanford University. K.M. was supported by a Royal Academy of Engineering Chair in Emerging Technology, UK.

## Author contributions

M.B.B. performed surgeries; M.B.B. and N.M. performed in-vivo imaging, immunohistochemistry and confocal imaging; D.P.H., S.V.S., and M.B.B. conducted image processing and statistical analysis; M.B.B., N.M., and D.P.H. conducted the electrophysiological measurements; A.K.G. developed and edited the codes used for VEP signal processing; N.J. performed field mapping on all implants used; A.S., E.B., and L.G. fabricated subretinal implants under the guidance of K.M. and T.K.; R.D. sectioned the retina; D.P. guided the research and data analysis; all authors participated in writing and/or reviewing the paper.

## Competing interests

D.P. and T.K. serve as consultants for Science Corp. (formerly Pixium Vision). D.P.'s patents related to retinal prostheses are owned by Stanford University and licensed to Science Corp., the details of which are disclosed below. The remaining authors declare no competing interests. Patent:—patent applicant: Stanford University—name of inventor(s): D. Palanker, A. Vankov, M. Blumenkranz—application number: US 7,047,080—status of application: issued—specific aspect of manuscript covered in patent application: use of photovoltaic pixels for retinal prosthetics.
