## [Transparent Peer Review file · Nature Communications]

Enhancing Prosthetic Vision by Upgrade of a Subretinal Photovoltaic Implant in situ

Corresponding Author: Dr Mohajeet Bhuckory

Version 0:

Reviewer comments:

Reviewer #1

(Remarks to the Author)

Retinal implants were shown to restore vision but the provided visual acuity remains low. Novel implant structures were proposed to enhance the visual performance of the patients. This paper proposes to investigate on one hand the feasibility of removing a planar or a 3D implant, and on a second hand, the possibility to functionally exchange two planar implants. The main question is to assess whether such a double surgery would impact the expected visual performance such that implanted patients could be considered for an implantation with an improved second generation of implants.

Figures 4 and 5 illustrate the removal of a planar implant, a honeycomb implant or a pillar implant. However, in all these cases, no statistical analysis is provided. Therefore, it appears difficult to decide whether the presented illustrations are representative or the best example.

Furthermore, additional immunostaining would be valuable to examine how the tissue structure is permanently affected by the 3D structure. Are the columns devoid of cells across the INL representing the pillar positions (see the histological sections) ? How the neurites of horizontal cells organize around these empty scars? The GFAP staining could be quantified in the INL-IPL to avoid the predominant OLM and ILM staining. This strategy would enable the authors to define if the 3D structure generate a long-standing greater glial reaction.

Figure 7 illustrates the performance of the animals after the exchange from the prima implant to the smaller pixel planar device. It is nice to see the maintained VEP values. To limit the influence of animal variability in the measures, considering the percentage of the measure with respect to the maximum amplitude may restrict its influence ? Is it possible to add also the animals with the primary MP20 in fig 7F ?

In the discussion, the sentence (line 232) is overstating because 3D implants were not replaced by the MP20. The replacement was only achieved for the PRIMA implants. It does remain unclear if 3D shaped implants affect the tissue function at their removal, the data only provide a structural stability. Replacing 3D implants with MP20 with the visual acuity measurement would provide a functional assessment of the functional integrity of the tissue.

Line 232

Our study has shown the feasibility of safely removing devices of all 3 configurations, demonstrated reimplantation of the next-generation arrays, and validated the subsequent improvement in their performance.

Discussing the limitations of the provided data would be more interesting than simply repeating the data. It could open a real discussion on the perspectives of the study.

Reviewer #2

(Remarks to the Author)

The manuscript submitted by Bhuckory et al. describes enhancement of the first-generation subretinal photovoltaic implant (PRIMA) to improve resolution from 100um to 28um. Data presented discuss changes to the device, surgical protocols for removal and replacement of an existing device, and some post-removal assessments of tissue integration for first generation devices. The implications are significant, as first-generation devices are currently in clinical trials and replacement of existing devices is a key obstacle to translating device upgrades to end-users. While the implications for device improvement are significant and the surgical procedure for replacement is a critical component of realizing these advances, the data presented for assessment of tissue integration and functional outcomes of the new devices data provide only

superficial assessment of retinal integration, integrity, and function.

Primary concerns and recommendations:

1. Integration of devices with the retina is a critical aspect of long-term safety and efficacy. Figure 2 depicts 3 geometries for subretinal arrays and attempts to illustrate tissue integration. All micrographs are displayed as rendered images. The corresponding confocal micrographs must be provided for comparison to illustrate the accuracy of rendering and allow readers to assess validity of claims.
2. Claim of cell type specificity with DAPI labeling is inappropriate in Figure 2. DAPI could be Muller glia, horizontal cell, bipolar cells, microglia, etc. If cell identity is claimed, this should be identified with the appropriate cell type specific markers. Feasibility of doing so is confirmed by IHC staining presented in other figures.
3. Assessments of debris depicted in Figure 3 are incomplete. Based on data regarding fibrosis and Muller glia, assessment of these components and any contribution debris are necessary. This is easily achieved by histological stain and IHC.
4. Assessment of only "reactive" Muller glia is insufficient. Based on histological data presented in Figure 5, there are some concerns regarding integrity of the INL and structural implication to Muller glia. The Muller glia population at-large should be assessed by more stable cell type-specific markers as well, i.e. glutamate transporters.
5. Inclusion of Iba-1 in histological data presented in Figure 5 would be valuable to support claims of microglia infiltration and interaction with implants presented in Figure 3.
6. VEP is not the ideal metric for demonstrating retinal activity promoted by the implant. VEPs assess cortical responses. The caveats of VEP interpretation are well-established, particularly activity arising from other cortical connectivity and variations in electrode placement. In this case, ERGs that include the photopic negative response (RGC-specific) are the most appropriate outcomes for directly demonstrating the electrophysiological output of the retina.
7. Assessment of hyper-reflective fibrosis and its mediators is lacking. Visualization of Muller glia and microglia within the implant, in retina during implantation, and after re-implantation would provide much needed insight on how these cells and their interaction with both first- and next-generation devices. Most of the improvements suggested in points 1-6 would provide at least some of the additional data to determine the origin of this fibrosis with greater certainty.

Version 1:

Reviewer comments:

Reviewer #1

(Remarks to the Author)

The authors have addressed my concerns.

All my thanks

Serge

Reviewer #2

(Remarks to the Author)

The manuscript submitted by Bhuckory et al. describes enhancement of the first-generation subretinal photovoltaic implant (PRIMA) to improve resolution from 100um to 28um. Data presented discuss changes to the device, surgical protocols for removal and replacement of an existing device, and some post-removal assessments of tissue integration for first generation devices. The implications are significant, as first-generation devices are currently in clinical trials and replacement of existing devices is a key obstacle to translating device upgrades to end-users. The implications for device improvement are significant and the surgical procedure for replacement is a critical component of realizing these advances. Revisions to the manuscript improve clarity of data reporting and provide a more comprehensive assessment of retinal integration and integrity.

We would like to thank the reviewers for carefully reading the manuscript, and for their questions and comments. We addressed them below (italicized) and made corresponding changes in the manuscript (tracked changes).

Reviewer #1:

Figures 4 and 5 illustrate the removal of a planar implant, a honeycomb implant or a pillar implant. However, in all these cases, no statistical analysis is provided. Therefore, it appears difficult to decide whether the presented illustrations are representative or the best example.

We agree and have added statistical analyses (Fig. 4 and supplementary Fig. 1) to help understand the trends across the sample size.

Furthermore, additional immunostaining would be valuable to examine how the tissue structure is permanently affected by the 3D structure. Are the columns devoid of cells across the INL representing the pillar positions (see the histological sections) ? How the neurites of horizontal cells organize around these empty scars? The GFAP staining could be quantified in the INL-IPL to avoid the predominant OLM and ILM staining. This strategy would enable the authors to define if the 3D structure generate a long-standing greater glial reaction.

The columns devoid of cells could be a result of the dynamic nature of the INL with cellular migration after the initial implantation and again after the extraction. While there is no way of actually knowing whether there was a pillar in that exact spot (due to rearrangement of INL cells after extraction of implants), the size of the columns devoid of cells neither correspond to the size of pillars (13-17 μm), nor the penetration depth of the 35 μm tall pillars. These columns could have been exacerbated by the sectioning technique (frozen sections fixed in 4% PFA) as no columns are seen in Fig 5 A and H (epoxy sections fixed in glutaraldehyde).

We have added Muller cell (GS) to figure 5, and for space reasons; and microglial (IBA1), horizontal (calbindin) cell markers to Supplementary Figure 1. We have now quantified the GFAP staining from INL-IPL to avoid the Muller cell endfeet and astrocytes in the layers beyond the ganglion cell layer. Similarly, we have quantified microglial cells within the INL and subretinal space. The quantification graphs were added to Supplementary figure 1.

Figure 7 illustrates the performance of the animals after the exchange from the prima implant to the smaller pixel planar device. It is nice to see the maintained VEP values. To limit the influence of animal variability in the measures, considering the percentage of the measure with respect to the maximum amplitude may restrict its influence? Is it possible to add also the animals with the primary MP22 in fig 7F?

Response to alternating gratings with primary MP22 implants was published in [15] (Figure 7b, plotted in units of spatial frequency (1/ (grating bar width)), and in units of bar width – in Supplementary Figure S6 b. We now added a citation of that publication in this sentence:

“The logarithmic fitting curve crossed the noise level at the 28 μm mark (Fig. 7F), matching the natural limit of visual acuity in rats [19], [20] and our previous results with primary MP22 implants [15].”

We have added primary MP22 acuity measurements to Fig. 7F:

In the discussion, the sentence (line 232) is overstating because 3D implants were not replaced by the MP20. The replacement was only achieved for the PRIMA implants. It does remain unclear if 3D shaped implants affect the tissue function at their removal, the data only provide a structural stability. Replacing 3D implants with MP20 with the visual acuity measurement would provide a functional assessment of the functional integrity of the tissue. Discussing the limitations of the provided data would be more interesting than simply repeating the data. It could open a real discussion on the perspectives of the study.

We now rephrased that sentence to more accurately describe the conclusions of the study: “Our study has shown the feasibility of safely removing devices of all 3 configurations. We also demonstrated reimplantation of the next-generation arrays instead of the first-generation flat PRIMA implants and validated the subsequent improvement in spatial resolution.”

Reviewer #2:

1. Integration of devices with the retina is a critical aspect of long-term safety and efficacy. Figure 2 depicts 3 geometries for subretinal arrays and attempts to illustrate tissue integration. All micrographs are displayed as rendered images. The corresponding confocal micrographs must be provided for comparison to illustrate the accuracy of rendering and allow readers to assess validity of claims.

We have added the full unrendered top view, as requested, to figure 2 (G-I), in addition to the rendered side views. All images were taken at the same height above the implant and provide a full view of the integration of each geometry with the retina.

2. Claim of cell type specificity with DAPI labeling is inappropriate in Figure 2. DAPI could be Muller glia, horizontal cell, bipolar cells, microglia, etc. If cell identity is claimed, this should be identified with the appropriate cell type specific markers. Feasibility of doing so is confirmed by IHC staining presented in other figures.

We removed cell specific identity claims. Where cell identify is suggested, we made sure to reference our cell identity migration papers:

M. B. Bhuckory *et al.*, “Cellular migration into a subretinal honeycomb-shaped prosthesis for high-resolution prosthetic vision,” *PNAS.*, vol. 120, no. 42, p. e2307380120, Oct. 2023, doi: 10.1073/pnas.2307380120.

M. Bhuckory *et al.*, “3D electronic implants in subretinal space: long-term follow-up in rodents,” *Biomaterials* (2024)

3. Assessments of debris depicted in Figure 3 are incomplete. Based on data regarding fibrosis and Muller glia, assessment of these components and any contribution debris are necessary. This is easily achieved by histological stain and IHC.

Figure 3 depicts biological remnants on the implants at the time of explantation. Our aim was to show that the biological material on the implant are immune cells (as shown by IBA1 staining) that have attached to the implant, rather than parts of the retina that were torn off during extraction. We have removed the term ‘debris’ from Figure 3 legend and discussion to avoid the confusion with the term ‘debris’ as used in the rest of the paper to describe a subretinal acellular layer present in AMD patients and observed in RCS rats after implant removal (Fig 5). This debris is believed to come from the RPE side (below the implant) and would not be present on top of the implants extracted in Fig. 3.

Figure 3 legend was modified as followed:

“Figure 3 demonstrates residual biological material on surface of the implants of all 3 geometries compared to pristine pre-implanted devices (Fig. 2A-C). Cell-shaped structures with rounded cell bodies and extending dendrite-like structures were observed in the higher magnification SEM images (red arrows; Fig. 3D-F). Acellular residues (yellow arrows) were also observed on the planar surface, inside the HC wells and between the pillars...”

4. Assessment of only “reactive” Muller glia is insufficient. Based on histological data presented in Figure 5, there are some concerns regarding integrity of the INL and structural implication to Muller glia. The Muller glia population at-large should be assessed by more stable cell type-specific markers as well, i.e. glutamate transporters.

We have now included glutamine synthetase (GS) staining to Figure 5 to a better picture of the Muller cell population at large.

5. Inclusion of Iba-1 in histological data presented in Figure 5 would be valuable to support claims of microglia infiltration and interaction with implants presented in Figure 3.

We have included IBA1 staining and have quantified IBA within the INL and subretinal space to assess the immune cell population present (Supplementary figure 1). There was no statistical significance between the control and the extraction groups. It is important to note that the immune cell infiltration seen in figure 3 on top of the implants is on the day of extraction itself (Day 0). Figure 5 illustrates the retina after 6 weeks (Day 42) of recovery from the extraction surgery. Since the foreign body (implant) is not present anymore in the subretinal space, microglial cells having a peak activation and infiltration of about 5-7 days were not expected to be abnormally present after 6 weeks.

6. VEP is not the ideal metric for demonstrating retinal activity promoted by the implant. VEPs assess cortical responses. The caveats of VEP interpretation are well-established, particularly activity arising from other cortical connectivity and variations in electrode placement. In this case, ERGs that include the photopic negative response (RGC-specific) are the most appropriate outcomes for directly demonstrating the electrophysiological output of the retina.

Due to the very strong artifact of subretinal electrical stimulation in ERG recordings, discerning the cellular response is quite difficult. In VEP, the artifact is much smaller, and can be easily removed (or even ignored), thus providing a clean physiological signal. An example ERG and its corresponding VEP signal in the same animal in response to prosthetic stimulation is provided below:

7. Assessment of hyper-reflective fibrosis and its mediators is lacking. Visualization of Muller glia and microglia within the implant, in retina during implantation, and after re-implantation would provide much needed insight on how these cells and their interaction with both first- and next-generation devices. Most of the improvements suggested in points 1-6 would provide at least some of the additional data to determine the origin of this fibrosis with greater certainty.

The aim of the current study was the assessment of feasibility of safe removal of subretinal implants, mainly due to FDA inquiries about this issue. Therefore, the goal here was the evaluation of preservation of the INL and more interior layers.

Our next publication is looking into the mediators of subretinal fibrosis in presence of subretinal devices and how to modulate them. The origin of this fibrosis is thought to be from RPE/choroid contribution. This could explain why no fibrosis is observed when re-implantation of a new device is performed immediately.

Our photovoltaic implants do not allow light to penetrate through and thus it is not possible to OCT the back of the implants. We have fabricated SU8 discs to mimic implants and study the RPE and choroid behind. We observed that the hyper-reflective layer is present behind the implant prior to explantation: